# Sub-Immunosuppressive Tacrolimus Ameliorates Amyloid-Beta and Tau Pathology in 3xTg-AD Mice

**DOI:** 10.3390/ijms26051797

**Published:** 2025-02-20

**Authors:** Jacqueline Silva, Batbayar Tumurbaatar, Jutatip Guptarak, Wen-Ru Zhang, Anna Fracassi, Giulio Taglialatela

**Affiliations:** 1Mitchell Center for Neurodegenerative Disease, Department of Neurology, The University of Texas Medical Branch at Galveston, Galveston, TX 77550, USA; jacsilva@utmb.edu (J.S.); batumurb@utmb.edu (B.T.); juguptat@utmb.edu (J.G.); wezhang@utmb.edu (W.-R.Z.); anfracas@utmb.edu (A.F.); 2The Moody Brain Health Institute, The University of Texas Medical Branch at Galveston, Galveston, TX 77550, USA

**Keywords:** tacrolimus, FK506, calcineurin, Alzheimer’s disease, 3xTg-AD, amyloid-beta, tau

## Abstract

Tacrolimus (TAC) has emerged as a potential therapy for Alzheimer’s disease (AD), with the challenge of balancing its therapeutic benefits against its immunosuppressive effects. This study explores the efficacy of a sub-immunosuppressive TAC dosing regimen to ameliorate AD-related pathologies. TAC was administered daily for 14 days, with drug concentrations measured via liquid chromatography tandem mass spectrometry (LC-MS/MS) in whole blood and hippocampal tissue from C57BL6J mice, while immunofluorescence analyses and Western blotting (performed on hippocampal extracts) were conducted in 10–12 month old 3xTg-AD mice to evaluate levels of tau and amyloid-beta (Aβ) proteins. The results from LC-MS/MS revealed that lower TAC doses resulted in sub-immunosuppressive blood levels, while still penetrating the hippocampi. Immunofluorescence showed reductions in tau and Aβ proteins in 3xTg-AD mice. Additionally, Western blot analyses revealed reductions in tau and Aβ, along with increases in synaptic and autophagy-related proteins. These findings highlight the potential of sub-immunosuppressive TAC doses in effectively targeting AD pathology while minimizing the risk of chronic systemic immunosuppression. Further research and clinical trials are warranted to establish the optimal TAC dosing regimen for AD treatment.

## 1. Introduction

Alzheimer’s disease (AD) is a multifaceted neurodegenerative disorder characterized by an accumulation of amyloid-beta (Aβ) and tau proteins, as well as early neurodegeneration in the hippocampus, a key brain region for learning and memory [1]. The hippocampus is further subdivided into regions, including the dentate gyrus (DG), cornu ammonis 1 (CA1), and cornu ammonis 3 (CA3). Each of these subregions is variably affected by hallmark AD pathologies, such as neurofibrillary tangles (NFTs) composed of misfolded and/or hyperphosphorylated tau and Aβ plaques [1]. Damage to these specific areas is central to the cognitive deficits observed in AD.

Tacrolimus (TAC), an FDA-approved immunosuppressant, has garnered significant attention as a potential therapy for the treatment and prevention of AD [2]. Several preclinical studies have demonstrated that TAC can reduce hallmark AD pathologies. For example, in transgenic mouse models of AD, TAC treatment reduces Aβ plaque burden and decreases tau hyperphosphorylation, improving memory deficits and restoring synaptic integrity [3,4,5,6,7]. Additionally, TAC has been reported to protect against age-associated microstructural changes in the beagle brain [8]. Complementing these preclinical findings, an observational study in human patients suggested a reduced prevalence of dementia in patients prescribed TAC (or related immunosuppressants), a conclusion further discussed in recent commentary [9,10]. Although these positive effects highlight the potential of TAC to mitigate AD pathology, they have been predominantly observed at doses that also induce systemic immunosuppression.

To overcome this limitation, a novel approach is to harness the benefits of TAC without inducing its immunosuppressive properties by employing a sub-immunosuppressive dosing regimen. Central to TAC’s therapeutic potential in AD is its mechanism of action as a calcineurin (CaN) inhibitor. CaN, a phosphatase involved in the dephosphorylation of various proteins, plays a pivotal role in cellular signaling pathways. Importantly, TAC’s lipophilic nature facilitates its passage across the blood-brain barrier (BBB), ensuring that it reaches central nervous system targets where it can modulate neuronal signaling [11]. By inhibiting calcineurin, TAC modulates key processes implicated in AD pathology, such as the activity of cyclic AMP-responsive element-binding protein (CREB) and autophagy mechanisms [12,13]. CREB, a transcription factor crucial for neuronal plasticity and survival, and autophagy, a cellular degradation pathway important for clearing misfolded proteins, represent potential targets through which TAC could exert neuroprotective effects and mitigate the accumulation of Aβ and pathological tau proteins, which are hallmark features of AD [12,13,14].

Despite these promising neuroprotective effects, the therapeutic use of TAC in AD is challenged by its dose-dependent immunosuppressive effects. Given that AD predominantly affects the elderly, who are already immunologically vulnerable, such immunosuppression could leave patients at an increased risk of infections and other complications. This duality, where the beneficial effects on AD pathology are counterbalanced by compromised immune function, exemplifies the principle articulated by Paracelsus, “the dose makes the poison” [15]. Any therapeutic strategy must avoid additional insults, making it imperative to establish an optimal dosing regimen that preserves the neuroprotective effects of TAC without inducing adverse immunosuppression.

In this context, the 3xTg-AD mouse model, genetically engineered to express three mutated human genes associated with AD—amyloid precursor protein (APP), microtubule-associated protein tau (tau), and presenilin 1 (PSEN1)—serves as a valuable tool [16]. These alterations in the 3xTg-AD model replicate the key pathological hallmarks of human AD, providing a comprehensive platform for studying the disease and evaluating therapeutic interventions. Using preclinical models, we aim to assess the efficacy of TAC at a potentially sub-immunosuppressive dose. Our goal is to target AD-related pathologies while minimizing the risk of chronic systemic immunosuppression, establishing a viable and safer method for TAC use in AD.

## 2. Results

### 2.1. Tacrolimus Concentration

Our LC-MS/MS analysis conducted in C57BL/6J mice revealed distinct TAC concentrations in whole blood and brain tissues of treated mice. Pooled blood samples from C57BL/6J mice treated with the lower TAC dose (0.1 mg/kg) contained a concentration of 2.96 ng/mL. Pooled blood samples from C57BL/6J mice receiving the higher dose (1.0 mg/kg) contained a concentration of 27.97 ng/mL of TAC (Figure 1A).

Furthermore, TAC was successfully detected in perfused hippocampal tissue from C57BL/6J mice receiving either dose. Six hours after the final TAC administration, the hippocampal tissue from mice treated with the low dose contained an average TAC concentration of 0.12 ng/g. Correspondingly, mice treated with the higher dose contained a higher average concentration of 2.7 ng/g in their hippocampal tissue (Figure 1B).

### 2.2. Protein Analysis in 3xTg-AD Mice

Our immunofluorescence studies revealed notable differences in tau and Aβ levels in the hippocampal regions of 3xTg-AD mice treated with TAC (0.1 mg/kg, i.p.) daily for 14 days compared to those receiving the vehicle. Mice treated with TAC exhibited reduced phosphorylated tau protein levels across all examined regions of the hippocampus, including the CA1, CA3, and DG (Figure 2, Figure 3 and Figure 4). Additionally, a reduction in Aβ was observed in the CA1 region of the hippocampus in TAC-treated mice compared to vehicle-treated control mice (Figure 2). However, in the CA3 and DG regions, the differences in Aβ levels between the groups did not achieve statistical significance (Figure 3 and Figure 4). When analyzing all three hippocampal regions collectively, there was an overall reduction in both phosphorylated tau and Aβ levels in the TAC-treated mice relative to the vehicle-treated mice (Appendix A).

Our Western blot analyses demonstrated that 3xTg-AD mice treated with 0.1 mg/kg TAC i.p. exhibited lower levels of tau and phosphorylated tau compared to vehicle-treated mice (Figure 5A–D). Additionally, the TAC treatment resulted in increased levels of phosphorylated-CREB, PSD95, and α-actinin proteins (Figure 5E–G). Surprisingly, western blot analyses indicated an increase in phosphorylated-mTOR and a decrease in phosphorylated p70S6K in the mice treated with TAC (Figure 6A–B). The examination of a panel of autophagy-related proteins showed elevated levels of p62, LC3, Atg5, and Atg7 in TAC-treated mice compared to those treated with the vehicle, with no significant changes observed in the levels of beclin-1, Atg16 L1, or Atg3 proteins (Figure 6C–I).

## 3. Discussion

In our investigation of TAC used for AD treatment, we focused on fine-tuning the dosing regimen to balance its neuroprotective benefits with minimal risk of systemic immunosuppression. Traditionally, TAC is used in transplant medicine to prevent graft rejection, with a therapeutic range of 5–20 ng/mL in blood associated with immunosuppressive effects [17,18]. Our objective was to explore whether TAC could be repurposed for AD treatment at sub-immunosuppressive doses, avoiding the chronic systemic immunosuppression associated with its standard clinical use.

We have previously demonstrated that administering TAC at a higher dose (1 mg/kg, i.p.) for 14 days reduces tau and Aβ proteins in the hippocampus of 3xTg-AD mice [19]. Building on these findings, we aimed to determine the blood concentration resulting from this dosing regimen to assess the likelihood of systemic side effects, particularly immunosuppression. Our results indicate that TAC at a dose of 1 mg/kg results in blood levels of 27.97 ng/mL, indicative of immunosuppression. Consequently, we explored whether a lower dose could effectively reduce AD-related pathology without the risk of chronic systemic immunosuppression.

Given that the established therapeutic range for effective immunosuppression with TAC requires blood trough levels above 3 ng/mL, the lower dose used in our study, which resulted in TAC levels below this threshold, can be considered sub-immunosuppressive [17]. Notably, blood trough levels of TAC are typically measured 12 h post-administration [17]. Our samples were taken 6 h after dosing, suggesting that the actual trough levels may be even lower, further distancing them from immunosuppressive concentrations. This was a crucial observation, as our aim was to investigate the potential neuroprotective effects of TAC without inducing chronic systemic immunosuppression. Since chronic systemic immunosuppression has been a major concern in the repurposing of TAC for use in AD patients, our study indicates the potential for safely using lower doses of TAC to target AD pathology. This approach mitigates the risks associated with higher immunosuppressive doses typically used in transplant medicine.

Our analysis focused on whole blood rather than plasma to align with the established guidelines for TAC measurement [17]. This method allows for comparison with the established therapeutic range for TAC-induced immunosuppression, which is based on whole blood trough concentrations. Such an approach is essential to accurately assess the risk of immunosuppression, a critical consideration for employing TAC as a therapeutic to treat neurodegenerative diseases. By adhering to clinical measurement standards, we ensure that our findings on TAC’s effectiveness and safety in neurodegenerative disease treatment are both practical and reliable.

After establishing a sub-immunosuppressive dose of TAC that effectively penetrates brain tissue, as confirmed by LC-MS/MS, our next objective was to evaluate the efficacy of this dose in mitigating AD-related pathology. Specifically, we assessed the impact of a sub-immunosuppressive TAC dose on the levels of phosphorylated-tau and Aβ proteins in the hippocampus of 3xTg-AD mice. This step was essential to evaluate whether the identified TAC dose could offer therapeutic benefits in the context of AD, without the risk of systemic immunosuppression. Our immunofluorescence results revealed that TAC (0.1 mg/kg, i.p.) was effective in reversing the AD-related pathology accumulated in the aged 3xTg-AD mice. Using integrated density measurements, we quantified levels of degradation-resistant paired helical filament (PHF) phosphorylated-tau and Aβ, key pathological hallmarks of AD [20,21]. Our results revealed that, compared to the vehicle treatment, a sub-immunosuppressive dose of TAC reduced the levels of phosphorylated-tau in all three investigated hippocampal regions. Although individual analysis of hippocampal subregions showed that Aβ levels were significantly reduced only in the CA1 region, collective analysis across the three regions indicated an overall decrease in both phosphorylated tau and Aβ levels in TAC-treated 3xTg-AD mice relative to vehicle controls. This regional discrepancy in Aβ reduction may reflect differences in regional susceptibility to Aβ pathology. Further investigation is needed to elucidate the underlying causes of these region-specific effects. Our findings suggest that a sub-immunosuppressive TAC dose holds potential to mitigate both tau and Aβ pathologies in the context of AD while minimizing the risks associated with systemic immunosuppression.

In addition to the observed effects on TAC concentration, Aβ, and tau pathologies, Western blot analyses have shed light on the molecular underpinnings of TAC’s neuroprotective effects. Treatment with 0.1 mg/kg TAC resulted in a reduction of total tau proteins. When analyzed separately, we observed decreases in tau monomers and tau oligomers, the latter of which are considered the most toxic tau species. Importantly, TAC treatment resulted in decreased levels of phosphorylated tau, consistent with our immunofluorescence results. This finding highlights that CaN is not directly responsible for the dephosphorylation of tau. In fact, CaN is known to dephosphorylate glycogen synthase-kinase-3b (GSK-3b), rendering it active [22]. Activated GSK-3b leads to tau phosphorylation, promoting the formation of NFTs [23]. Therefore, TAC’s inhibition of CaN likely results in GSK-3b remaining phosphorylated and thus inactive, hindering tau phosphorylation and leading to the observed reduction in phosphorylated tau levels.

Treatment with TAC at a sub-immunosuppressive dose resulted in increased levels of synaptic integrity markers, PSD95 and α-actinin, in the 3xTg-AD mice, indicating enhanced synaptic stability. This enhancement is bolstered by the increase in phosphorylated CREB in the TAC-treated group, which suggests a reactivation of synaptic activity, as CREB activation is known to drive the transcription of genes critical for synaptic function [24]. The observed enhancements in synaptic integrity markers, coupled with the increase in phosphorylated CREB, suggest a reactivation of synaptic activity and potential remodeling. This is critical, as it indicates not merely a symptomatic alleviation but a fundamental alteration in the disease’s trajectory, positioning TAC as a candidate for truly disease-modifying treatment for AD.

Given that AD pathology is known to dysregulate autophagic pathways, and based on our previous observations that CaN inhibition can reinitiate autophagy, we sought to assess whether TAC treatment at the sub-immunosuppressive dose modulates key markers of this degradation process [7,25]. TAC treatment modulated autophagy pathways, as evidenced by increased levels of p62, Atg5, Atg7, and LC3, indicating an activation of cellular clearance mechanisms. We measured mTOR expression as a marker of autophagy, given its central role in regulating this process. Unexpectedly, TAC treatment led to an increase in phosphorylated mTOR, alongside a reduction in phosphorylated p70S6K, the most immediate downstream mTOR target. This observation suggests a nuanced interaction with the mTOR pathway that diverges from traditional autophagy regulation. It may reflect differences in compensatory responses or direct effects of TAC on cellular signaling. Further studies—including assessments of autophagic flux and the use of specific inhibitors or activators of autophagy and mTOR signaling—are necessary to fully elucidate these underlying mechanisms. Nonetheless, our findings underscore the potential of TAC to modulate autophagy, enhancing the degradation and clearance of pathogenic proteins and reinforcing its therapeutic potential as a disease-modifying treatment for AD.

These molecular alterations provide a deeper understanding of TAC’s multifaceted role in AD and lay the groundwork for future studies aimed at unraveling the precise mechanisms by which TAC confers neuroprotection. Importantly, we have identified a sub-immunosuppressive dose of TAC that not only reduces AD-related pathology but also induces synaptic stability in a preclinical AD model, potentially through modulating autophagy.

Further research is necessary to solidify the optimal dosing regimen for TAC that achieves sub-immunosuppressive blood concentrations and effectively addresses AD pathology without inducing chronic systemic immunosuppression in humans. Our study lays the groundwork for this endeavor, suggesting the feasibility of this approach. Building on these findings, we propose the initiation of a small-scale clinical trial involving patients diagnosed with early AD to evaluate the effects of sub-immunosuppressive doses of TAC in a clinical setting and translate these preclinical observations into potential disease-modifying therapeutic interventions for AD.

## 4. Materials and Methods

### 4.1. Animals

The University of Texas Medical Branch (UTMB) Institutional Animal Care and Use Committee (IACUC) approved all animal experiments. C57BL/6J (n = 8) and 3xTg-AD (n = 8) transgenic mice were obtained from Jackson Labs (Bar Harbor, ME, USA). The 3xTg-AD mice were maintained in the breeding program at UTMB. All mice used in this study were male. Mice were maintained at the UTMB vivarium under USDA standards (12 h light/dark cycle; food and water ad libitum). At the onset of the study, the C57BL/6J mice reached an age of 12 weeks, and the 3xTg-AD mice were aged between 10 and 12 months. All mice were subjected to a daily treatment regimen of either TAC or a vehicle control for 14 days. Mice were euthanized by exposure to isoflurane vapors followed by cervical dislocation 6 h after the last treatment. Whole blood was collected, then mice were perfused with phosphate buffered saline (PBS). Following perfusion, brains and other organs were rapidly dissected and frozen immediately on dry ice. Organs were then stored at -80 °C until further analyses were conducted. One hemisphere of the brain from each 3xTg-AD mouse was placed in 5 mL of 4% paraformaldehyde (PFA) at 4 °C for 48 h to prepare for immunofluorescence.

### 4.2. Tacrolimus Treatment

TAC (PROGRAF^®^, Astellas Pharma, Tokyo, Japan) was obtained from the UTMB pharmacy as a 5 mg/mL stock solution. Each day, this stock was diluted in filtered PBS to prepare fresh working solutions. For the 1.0 mg/kg dose, a 0.1 mg/mL working solution was administered via intraperitoneal (i.p.) injection at 10 µL per gram of body weight. For the 0.1 mg/kg dose, a 0.01 mg/mL working solution was similarly administered at 10 µL per gram of body weight. All mice received the same dose every 24 h for 14 consecutive days. Six hours after the final treatment on day 14, the mice were euthanized.

### 4.3. Liquid Chromatography Tandem Mass Spectrometry

Prior to conducting this study, an acute TAC exposure experiment was performed where mice received a single dose of TAC or VEH and were euthanized 6 h later. Whole blood was collected and liquid chromatography tandem mass spectrometry (LC-MS/MS) was performed. Since TAC was not detected in the VEH group, LC-MS/MS was not repeated in the VEH group for the current study (Appendix A). In the current study, two pooled whole blood samples and 8 hippocampal tissue samples from 8 C57BL/6J mice were sent to the UTMB Mass Spectrometry Facility for sample processing using LC-MS/MS. TAC was extracted by taking 50 µL of pooled whole blood and then combining it with 10 μL of 100 ng/mL isotopically labeled [^13^C-D_2_] TAC (FK-506-^13^C-d_2_, Cayman Chemical, ca. no. 22178) and 20 μL of 0.5 M ZnSO_4_ to lyse cells. Proteins were then precipitated with the addition of 500 μL of cold methanol. Samples were incubated at −80 °C for 30 min to aid in protein precipitation and then centrifuged to pellet protein and insoluble material. Supernatants were transferred to new tubes before drying under a stream of nitrogen gas. Dried samples were then reconstituted in 100 μL of LC mobile phase solvents (A/B, 5:95 v/v) before preparing them for LC-MS/MS analysis. For the extraction of TAC from brain tissue, approximately 20 mg of brain tissue was combined with 10 μL of 100 ng/mL [^13^C-D_2_] TAC and 300 μL of cold 80% methanol and then homogenized using silica beads. The homogenized samples were then extracted and reconstituted as described for the samples above. Control extractions included a solvent-only extraction to monitor for background signals and an internal standard (IS)-only extraction to determine extraction efficiency. Following extraction, an aliquot of each sample was pooled into a single vial to create a pooled QC, which was injected periodically throughout the LC-MS/MS queue to monitor signal and retention time stability of the TAC LCMS peaks. Calibration curves were created using unlabeled TAC normalized to [^13^C-D_2_] TAC prepared in matrix-matched backgrounds from concentrations of 0.5 to 500 ng/mL for blood and 0.25 to 25 ng/mL for brain (Appendix A). TAC and [^13^C-D_2_] TAC were also prepared as QC samples in matrix-matched backgrounds at known concentrations to validate the LCMS method.

LC-MS/MS was performed using an Acquity Premier UHPLC (Waters, Milford, MA, USA) coupled to a QTRAP 6500 mass spectrometer (SCIEX, Framingham, MA, USA). TAC was separated by reverse-phase chromatography on a BEH C18 column (50 × 2.1 mm, 1.7 μm, Waters) using mobile phases of: (A) water + 0.1% formic acid + 2 mM ammonium acetate and (B) acetonitrile + 0.1% formic acid + 2 mM ammonium acetate. The LC gradient was as follows: 30% B from 0 to 0.5 min, 30% B to 95% B from 0.5 to 1 min, 95% B from 1 to 2.6 min, 95% B to 30% B from 2.6 to 2.8 min, and 30% B from 2.8 to 4 min. The flow rate was set to 0.55 mL/min.

TAC and [^13^C-D_2_] TAC were detected using a multiple reaction monitoring (MRM) scan. The MRM transition for TAC was a precursor ion mass of 821.6 Da with a fragment ion mass of 768.3 Da using a declustering potential (DP) of 6 V and collision energy (CE) of 25 V. For the IS [^13^C-D_2_] TAC, the precursor ion mass was 824.7 Da with a fragment ion mass of 771.3 Da using a DP of 31 V and a CE of 35 V. The MS source parameters were the following: curtain gas set to 35 psi, collisional activated dissociation set to medium, ion spray voltage set to 5500 V, ion source temperature set to 500 °C, nebulizing gas (GS1) set to 50 psi, and heating gas (GS2) set to 55 psi. The instrument was operated and the data collected using Analyst software (v1.7.3, SCIEX).

Peak areas were integrated using either MultiQuant software (v3.0.3, SCIEX) or Skyline (v23.1.0.268, MacCoss Lab Software) and normalized to the peak area of the TAC [^13^C-D_2_] IS. Absolute quantification was performed by plotting the peak areas of standards against expected concentrations to obtain a linear regression equation. Concentrations of TAC in samples were calculated using the linear regression equation. Data were analyzed using Microsoft Excel and R/RStudio (R v4.3.1, RStudio v2023.09.1+494).

### 4.4. Immunofluorescence

After 48 h, brains from the 3xTg-AD mice were removed from PFA and placed in 30% sucrose for 7 days before being embedded in the O.T.C. compound (catalog# 4583, Tissue-Tek, Tokyo, Japan) and sectioned at 12 µm onto Superfrost/Plus slides (catalog# 12-550-15, Fisherbrand, ThermoFisher Scientific, Waltham, MA, USA). The prepared slides were stored at −80 °C until they were processed for immunofluorescence as described previously [25]. Slides were fixed with 4% PFA for 30 min at room temperature (RT). To block non-specific binding, 5% bovine serum albumin (Sigma-Aldrich, A4503-100G) and 10% normal goat serum (NGS) (Sigma-Aldrich, S26-100mL) were used, and sections were permeabilized with 0.5% Triton X-100 and 0.05% Tween-20 for an hour at RT. Primary antibodies, mouse anti-PHF13.6 (1:400, catalog# 35-5300, RRID:AB_2533211, Invitrogen), and rabbit anti-Aβ (1:200, catalog# ab201060, RRID:AB_2818982, Abcam, Cambridge, UK), which were diluted in PBS with 1.5% NGS, were applied overnight at 4 °C. Post-washing in PBS, slides were incubated with proper Alexa-conjugated secondary antibodies, goat anti-mouse Alexa Fluor 594 (1:400, catalog# A11032, RRID:AB_2534091, ThermoFisher Scientific), or goat anti-rabbit Alexa Fluor 488 (1:400, catalog# A11008, RRID:AB_143165, ThermoFisher Scientific) in PBS with 1.5% NGS for 1 h at RT. After additional PBS washes, slides were treated with 0.3% Sudan Black B in 70% ethanol for 10 min. Following a water rinse, slides were cover slipped with Fluoromount-G containing 4′,6-diamidino-2-phenylindole (DAPI, Statistical Methodology and Applications Branch 0100-20, SouthernBiotech, Birmingham, AL, USA) and sealed.

Immunoreacted sections were imaged using a Keyence BZ-X800 (Keyence Corporation, Osaka, Japan) microscope with a 40X objective. Three sections per animal were analyzed, with 7 images per section captured at 1920 × 1440 resolution with a z-step of 0.9 µm over a 12 µm thickness. For quantification, all layers of a single image were merged into one slice (stack/Z projection). Image J software (v1.54, Open Source from National Institutes of Health, Bethesda, MD) was used for quantitative analysis, measuring fluorescence intensity per area (Integrated Density) for each target. Representative images were compiled in Adobe Photoshop CC2023.

### 4.5. Western Blot Analysis

Western blot analyses were performed on total protein extracts from hippocampal tissue from 3xTg-AD mice treated with TAC (0.1 mg/kg, i.p.) or the vehicle (*n* = 4/group). Samples were processed by homogenization in 1X Radioimmunoprecipitation Assay (RIPA) buffer (catalog #9806, Cell Signaling Technology, Danvers, MA, USA) containing 1 mM phenylmethyl sulfonyl fluoride (PMSF) and 1X Halt Protease and Phosphatase Inhibitor Cocktail (catalog #78440, ThermoFisher Scientific). Samples were incubated for 30 min on ice for lysis, and tissue lysate was centrifuged for 15 min at 15,000× *g* at 4 °C. Protein concentrations were determined using the Pierce™ BCA Protein Assay Kit (catalog #23227, ThermoFisher Scientific, Waltham, MA, USA). Proteins were then separated by electrophoresis on 4–15% Mini-PROTEAN^®^ TGX™ Precast Protein Gels (catalog #4561086, Bio-Rad, Hercules, CA, USA) and transferred to Immobilon-P membranes (Millipore, St. Louis, MO, USA) for 1 h at 95 V at 4 °C. Membrane blocking was performed using 3% bovine serum albumin and Tris-buffered saline/0.1% Tween-20 (TBS-T) for 1 h at RT, followed by overnight incubation at 4 °C with primary antibodies: anti-tau13 (1:1000, catalog #835204, RRID:AB_2728543, BioLegend, San Diego, CA, USA), anti-phospho-tau (Ser202/Thr205; AT8) (1:1000, catalog # MN1020, RRID:AB_223647, Invitrogen, Carlsbad, CA, USA), anti-post synaptic density 95 (PSD95) (1:2000, catalog #3450, RRID:AB_2292883, Cell Signaling Technology, Danvers, MA, USA), anti-α-actinin (1:1000, catalog #3134, Cell Signaling Technology), anti-mammalian target of rapamycin (mTOR) (1:1000, catalog #2972, RRID:AB_330978, Cell Signaling Technology, Danvers, MA, USA), anti-phospho-mTOR (Ser2448) (1:1000, catalog #2971, RRID:AB_330970, Cell Signaling Technology, Danvers, MA, USA), anti-70-kDa ribosomal protein S6 kinase (p70S6K) (1:1000, catalog #2708, RRID:AB_390722, Cell Signaling Technology, Danvers, MA, USA), anti-phospho-p70S6K (Thr389) (1:1000, catalog# 9234, RRID:AB_2269803, Cell Signaling Technology, Danvers, MA, USA), anti-sequestosome-1 (p62) (1:1000, catalog #ab56416, RRID:AB_945626, Abcam), anti-cAMP response element binding (CREB) protein (1:1000, catalog #9104, RRID:AB_490881, Cell Signaling Technology, Danvers, MA, USA), anti-phospho-CREB (Ser133) (1:1000, catalog# 9196, RRID:AB_331275, Cell Signaling Technology, Danvers, MA, USA), autophagy antibody sampler kit including Beclin-1, Atg5, Atg16 L1, Atg7, Atg3, LC3 A/B (1:1000, catalog# 4445, Cell Signaling Technology, Danvers, MA, USA), and anti-β-actin (1:50000, catalog #A1978, RRID:AB_476692, Sigma-Aldrich, St. Louis, MO, USA). Following incubation, membranes were washed three times in 1X TBST for 10 min each, then incubated for 1 h at RT with HRP-conjugated secondary antibodies: anti-rabbit IgG (1:5000, catalog #7074, RRID:AB_2099233, Cell Signaling Technology, Danvers, MA, USA) and anti-mouse IgG (1:5000 catalog #7076, RRID:AB_330924, Cell Signaling Technology, Danvers, MA, USA). After secondary antibody incubation, the membranes underwent three 10 min washes. Signal detection was performed with Amersham Electrochemiluminescence Western Blotting Detection Reagents (catalog #RPN2209, Cytiva, Marlborough, MA, USA), and the density of the immunoreactive bands was quantified using ImageJ FIJI software (v.1.54, https://imagej.nih.gov/ij, National Institutes of Health, Bethesda, MD, USA).

### 4.6. Data Analysis

Statistical analyses were performed using GraphPad Prism version 10.1.1. One-tailed *t*-tests assessed differences between groups, with *p* < 0.05 being considered statistically significant. Data were graphed as means ± SEM using R/RStudio (R v4.3.1, RStudio v2023.09.1+494).

## 5. Conclusions

In conclusion, our study highlights the potential of TAC as a neuroprotective agent with disease-modifying potential for AD treatment. We demonstrate that a sub-immunosuppressive dose can effectively penetrate brain tissue and reduce key AD pathologies in the hippocampus. Crucially, this can be achieved without reaching blood levels associated with systemic immunosuppression, a major concern in repurposing TAC for AD. Our findings suggest that TAC, administered in carefully controlled doses, may offer a promising therapeutic avenue for AD. This study provides a foundation for further research and clinical trials to determine the optimal TAC dosing regimen for AD treatment, potentially transforming the management of this debilitating disease.

## Figures and Tables

**Figure 1 ijms-26-01797-f001:**
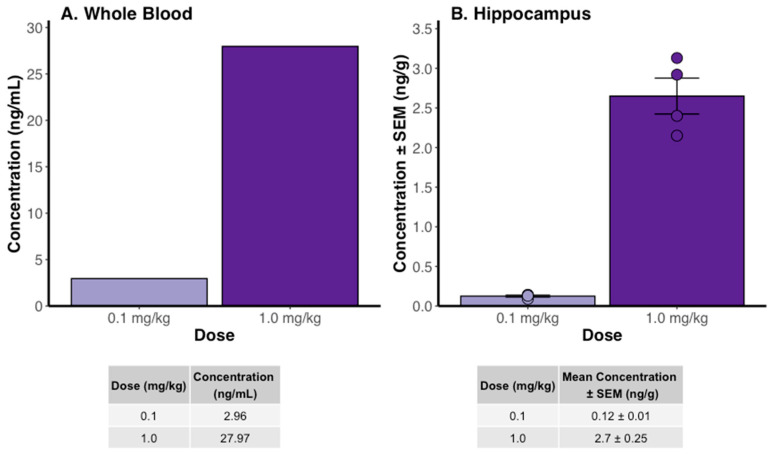
Tacrolimus Concentration. Tacrolimus (TAC) concentrations in (**A**) whole blood (*n* = 4 pooled samples/group) and (**B**) hippocampal tissue (*n* = 4/group) from C57BL/6J mice receiving TAC (0.1 mg/kg) or TAC (1.0 mg/kg) daily for 14 days.

**Figure 2 ijms-26-01797-f002:**
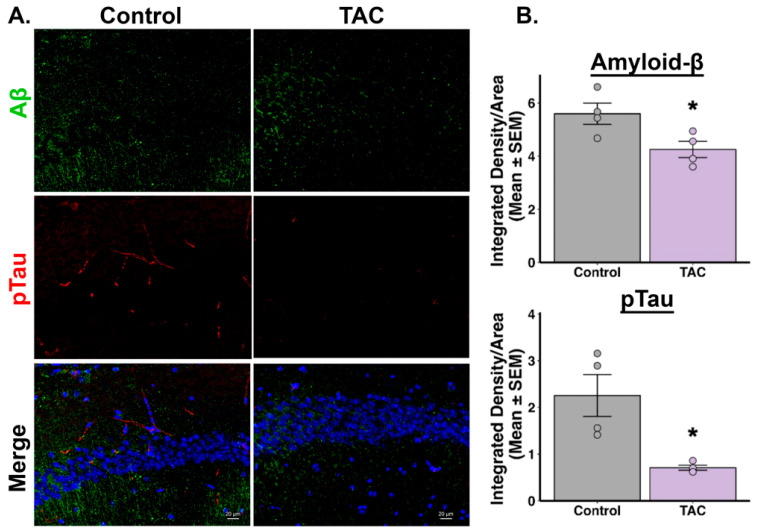
TAC reduces Aβ and tau protein levels in the CA1 region of 3xTg-AD Mice. (**A**) Representative images of Aβ and phosphorylated tau staining in the CA1 region of the hippocampus from TAC- and vehicle-treated 3xTg-AD mice (*n* = 4/group). (**B**) Quantitative analyses of integrated density reveal reduced levels of Aβ (* *p* = 0.018) and phosphorylated tau (* *p* = 0.007) in TAC-treated mice compared to vehicle-treated mice.

**Figure 3 ijms-26-01797-f003:**
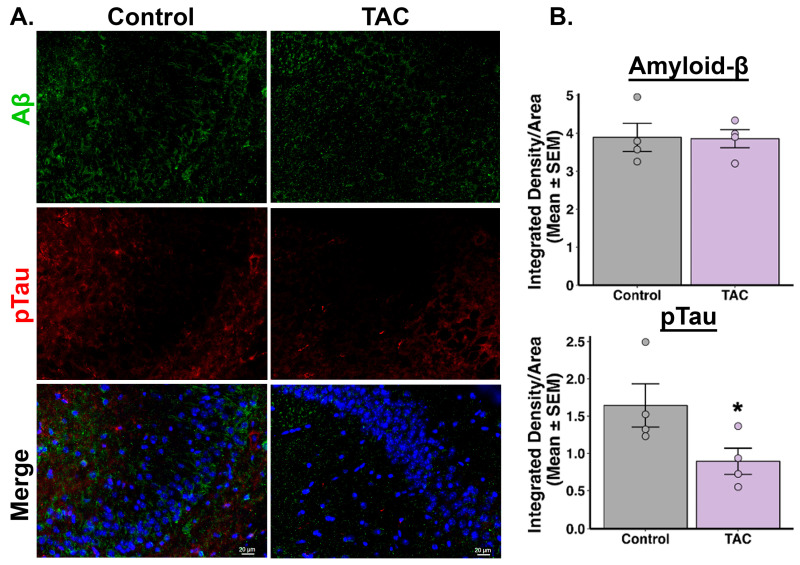
TAC reduces tau protein levels in the CA3 region of 3xTg-AD Mice. (**A**) Representative images of Aβ and phosphorylated tau staining in the CA3 region of the hippocampus from TAC- and vehicle-treated 3xTg-AD mice (n = 4/group). (**B**) Quantitative analyses of integrated density reveal no significant difference in Aβ (*p* = 0.469) levels and reduced levels of phosphorylated tau (* *p* = 0.035) in TAC-treated mice compared to vehicle-treated mice.

**Figure 4 ijms-26-01797-f004:**
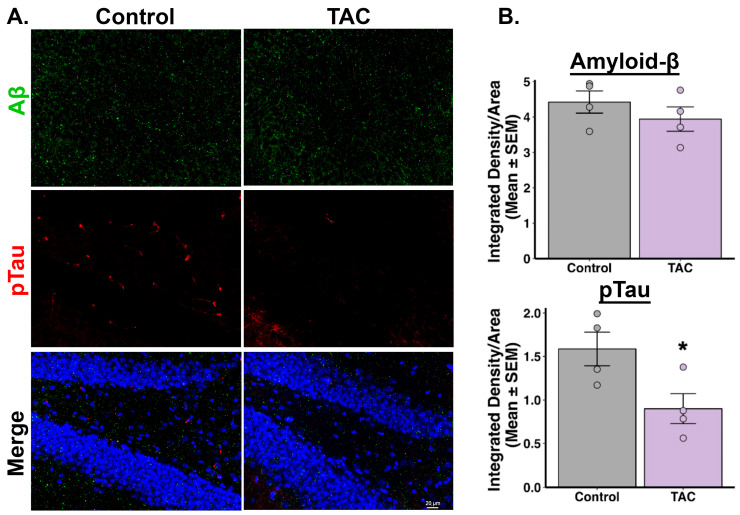
TAC reduces tau protein levels in the DG region of 3xTg-AD Mice. (**A**) Representative images of Aβ and phosphorylated tau staining in the DG region of the hippocampus from TAC- and vehicle-treated 3xTg-AD mice (n = 4/group). (**B**) Quantitative analyses of integrated density reveal no significant difference in Aβ (*p* = 0.171) levels and reduced levels of phosphorylated tau (* *p* = 0.019) in TAC-treated mice compared to vehicle-treated mice.

**Figure 5 ijms-26-01797-f005:**
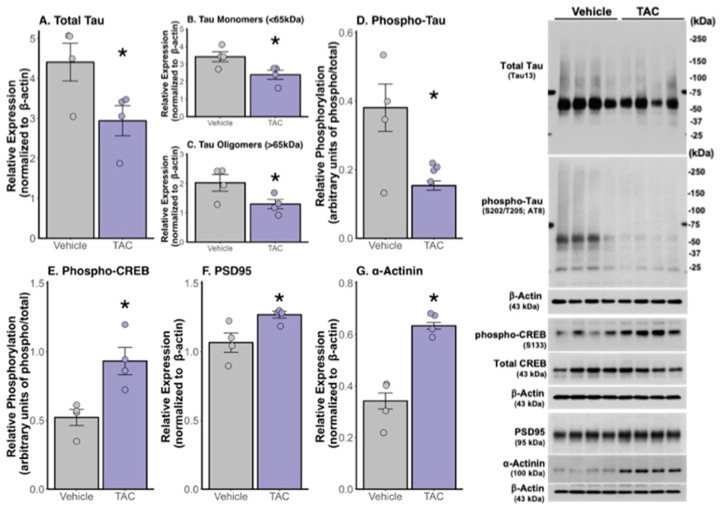
TAC reduces tau and increases synaptic protein levels in the hippocampus of 3xTg-AD mice. Western blot analyses of the total hippocampal tissue homogenate (*n* = 4/group) using the Tau13 antibody revealed decreases in (**A**) total tau (* *p* = 0.026), (**B**) tau monomers (* *p* = 0.019), and (**C**) tau oligomers (* *p* = 0.034). Western blot analyses also revealed a decrease in (**D**) phosphorylated-tau (* *p* = 0.009) and an increase in levels of synaptic proteins (**E**) phosphorylated CREB (* *p* = 0.006), (**F**) PSD95 (* *p* = 0.017), and (**G**) α-actinin (* *p* < 0.001) in TAC-treated mice (*n* = 4) compared to vehicle-treated mice (*n* = 4).

**Figure 6 ijms-26-01797-f006:**
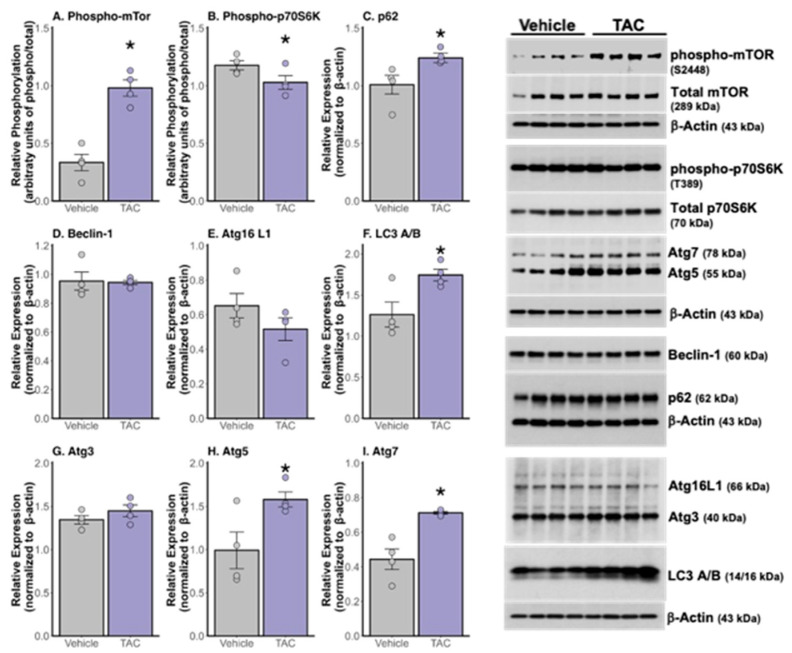
TAC increases autophagy-related protein levels in the hippocampus of 3xTg-AD mice. Western blot analyses of the total hippocampal tissue homogenate (n = 4/group) revealed an increase in (**A**) phosphorylated mTOR (* *p* = 0.0003), a decrease in (**B**) phosphorylated p70S6K (* *p* = 0.041), an increase in (**C**) p62 (* *p* = 0.022), no change in (**D**) beclin-1 or (**E**) Atg16 L1, an increase in (**F**) LC3 A/B (* *p* = 0.014), no change in (**G**) Atg3, and an increase in both (**H**) Atg5 (* *p* = 0.021) and (**I**) Atg7 (* *p* = 0.002) in TAC-treated mice (*n* = 4) compared to vehicle-treated mice (*n* = 4).

## Data Availability

Data is contained within the article and Appendix A.

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
