# Peer review of "Sub-Immunosuppressive Tacrolimus Ameliorates Amyloid-Beta and Tau Pathology in 3xTg-AD Mice"

_ijms, 2025, doi:10.3390/ijms26051797_

Round 1

Reviewer 1 Report

Comments and Suggestions for Authors

In the study “Sub-Immunosuppressive Tacrolimus Ameliorates Amyloid-Beta and Tau Pathology in 3xTg-AD Mice” by Silva et al., the authors explore the efficacy of a sub-immunosuppressive TAC dosing regimen to ameliorate AD-related pathologies.

The study is interesting but the MS needs to be restructured and explained in more detail in order to be comprehensive for the readers.

First of all, Figure 6 is missing.

The Abstract needs restructuring. For example: How old were the animals when treated? It should be mentioned in which structure were western blots performed?

Introduction:

Line 27: the authors state Tacrolimus (TAC), an FDA-approved immunosuppressant, has garnered significant attention as a potential therapeutic for the treatment and prevention of Alzheimer's disease (AD), supported by numerous studies that demonstrate its ability to alleviate AD-related pathology and behavioral deficits [1–9], and then continue to explain the mechanism of TAC action. It will be beneficial for readers to first briefly explain what are the positive effects of TAC in AD. Where these studies performed only in the preclinical models? This should be followed with the introduction of the novel approach presented in this study – the treatment with sub-immunosuppressive dose of TAC. Then, the TAC mechanism of action should be briefly explained.

Line 43: ….underscores the critical need to establish an optimal TAC dosing regimen that treats AD pathology without pre-cipitating chronic systemic immunosuppression [13]. This consideration is particularly crucial given the complex pathology of AD, which involves the accumulation of amyloid-beta () and tau proteins as well as early neurodegeneration in the hippocampus, a crucial area for learning and memory [14].   – please explain the causative between these two sentences? In general, what are the specific effects of high-dose TAC treatment in the AD mouse model? The authors gave 9 references indicating the positive effects of TAC treatment – it will be beneficial for readers if these effects are briefly explained in the MS.

The authors have published the paper in which they have used 1mg/kg TAC. In this study the authors use 1mg/kg and 0.1 mg/kg. In the Treatment section of Material and methods the doses are – 1.0 mg/ml and 0.1 mg/ml) and in the result section the doses are 1mg/kg and 0.1 mg/kg.  Please correct.

Results:

What is the number of animals used for each of the experiment? That should be stated in the Material and methods section of the MS and in the legend of every figure.

How come that there are individual measurements presented in the Fig.1B and not in the Fig1.A if both blood and hipp samples are pooled. Where is the statistics coming from?

It should be stated in the text (2.1 section of the Results) that the blood concentration of TAC was measured in WT mice.

Better representative images should be chosen for the Fig. 2A, Fig.3A, Fig.4A. for the p-Tau immunohistochemistry so that they can correspond to the graph values.

What is the rationale for not measuring potential changes in the amyloid plaque distribution using Th-S staining?

Considering the immunosuppressive effects of TAC what is the status of microglia and astrocyte activation i.e. GFAP and Iba1 expression? What is the status of inflammation markers Il1b, IL6, TNF-a?

Line 180: Additionally, this dose of TAC reduced Aβ levels, particularly in the CA1 region of the hippocampus.  – you mean ONLY in CA1 region. What is the explanation for this regional discrepancy? Please explain in more detail why was this particular antibody (Ab1-42) used. Are those Abeta monomers that were measured?

Line 208: Please give the rationale for the mTOR expression analyses.

The authors state that the sub-immunosuppressive 0.1mg/kg TAC treatment is more beneficial. However, in the paper “Chronic FK506 treatment reduces amyloid β and tau levels in the hippocampus and cortex of 3xTg‐AD mice”DOI:10.1002/alz.061984, the authors showed that 3xTG-AD mice treated with higher FK506 (i.e. TAC) dose (1mg/ml) treated mice displayed significant lower levels of Aβ and Tau in all the considered areas as compared to PBS‐treated 3xTg‐AD mice. How are these results compared to the results obtained in this study? Why is the dose of 0.1mg/kg more beneficial? Were there any documented negative side-effects when the higher dose was used?

The images are missing in the original images/gels folders.

Author Response

(1) First of all, Figure 6 is missing. We thank the reviewer for pointing out the absence of Figure 6. Please note that Figure 6 was included in our originally submitted manuscript; it appears that during the formatting process it was inadvertently omitted. We have now reinserted it in the revised version.

(2) The Abstract needs restructuring. For example: How old were the animals when treated? It should be mentioned in which structure were western blots performed? We thank the reviewer for the suggestion. The Abstract has been revised to include the animals' age at treatment and specify the structures used for Western blot analysis. Revisions are highlighted in tracked changes

(3) Line 27: the authors state… and then continue to explain the mechanism of TAC action. It will be beneficial for readers to first briefly explain what are the positive effects of TAC in AD. Where these studies performed only in the preclinical models? This should be followed with the introduction of the novel approach presented in this study–the treatment with sub-immunosuppressive dose of TAC. Then, the TAC mechanism of action should be briefly explained. We thank the reviewer for the valuable comment. In response, the introduction now first summarizes the positive effects of TAC in AD, explicitly noting that these studies were conducted in preclinical models, then introduces our novel approach using a sub-immunosuppressive dose of TAC, followed by a brief explanation of its mechanism of action. (Tracked changes are highlighted.)

(4) Line 43: ….please explain the causative between these two sentences? In general, what are the specific effects of high-dose TAC treatment in the AD mouse model? We have revised the text to clarify the link between the need for an optimal dosing regimen and the effects of high-dose TAC. Specifically, we now explain that while high doses of TAC in AD mouse models have reduced amyloid-beta and tau pathology, they also induce immunosuppression, underscoring the need for a dosing strategy that maximizes efficacy while minimizing unintended effects.

(5) The authors have published the paper in which they have used 1mg/kg TAC. In this study the authors use 1mg/kg and 0.1 mg/kg. In the Treatment section of Material and methods the doses are–1.0 mg/ml and 0.1 mg/ml) and in the result section the doses are 1mg/kg and 0.1 mg/kg.  Please correct. We thank the reviewer for pointing this out. In the revised manuscript, we have clarified that the values in mg/mL refer to the concentration of the TAC working solution, while the dosing (mg/kg) accounts for each animal’s weight. We have updated the Methods section to clearly explain this distinction.

(6) What is the number of animals used for each of the experiment? That should be stated in the Material and methods section of the MS and in the legend of every figure. We thank the reviewers for noting this oversight. The Materials and Methods section and all figure legends have now been updated to include the number of animals used in each experiment. (7) How come that there are individual measurements presented in the Fig.1B and not in the Fig1.A if both blood and hipp samples are pooled. Where is the statistics coming from? We thank the reviewer for highlighting this point. We clarify that blood samples were pooled for analysis, while individual hippocampal samples were analyzed, which is why individual measurements are only presented in Fig. 1B. The figure legends now include the appropriate n values to clarify the source of the statistics.

(8) It should be stated in the text (2.1 section of the Results) that the blood concentration of TAC was measured in WT mice. In Section 2.1 of the Results, we have clarified that the blood concentration of TAC was measured in C57BL/6J mice.

(9) Better representative images should be chosen for the Fig. 2A, Fig.3A, Fig.4A. for the p-Tau immunohistochemistry so that they can correspond to the graph values. We thank the reviewer for the suggestion. The images have been updated with more representative examples that accurately correspond to the graph values.

(10) What is the rationale for not measuring potential changes in the amyloid plaque distribution using Th-S staining? We thank the reviewer for this valuable comment. Th-S staining was not used because, while it selectively binds to β-sheet-rich amyloid fibrils, it does not detect diffuse or oligomeric amyloid species, components increasingly recognized as critical in AD pathology, especially at the presymptomatic stage examined in our study. Additionally, Th-S can cross-react with tau potentially confounding quantification (see Shin et al., Sci Rep, 11:1617, 2021). Our use of an anti-Aβ antibody allows for a more detailed and quantitative analysis of amyloid species. Moreover, significant plaque deposition typically occurs in older mice (with males often showing notable plaques ~15 months), further supporting our focus on immunofluorescence methods for this presymptomatic phase.

(11) Considering the immunosuppressive effects of TAC what is the status of microglia and astrocyte activation i.e. GFAP and Iba1 expression? What is the status of inflammation markers Il1b, IL6, TNF-a? We thank the reviewer for this interesting question. Since we used a sub-immunosuppressive dose of TAC, and the mice were not exposed to any pro-inflammatory challenge, we do not expect any significant changes in markers of microglia and astrocyte activation or in pro-inflammatory cytokine levels.

(12) Line 180: Additionally, this dose of TAC reduced Aβ levels, particularly in the CA1 region of the hippocampus.  – you mean ONLY in CA1 region. What is the explanation for this regional discrepancy? Please explain in more detail why was this particular antibody (Ab1-42) used. Are those Abeta monomers that were measured? While the reduction in Aβ levels reached statistical significance only in the CA1 region when analyzed separately, combined analysis of the three regions revealed an overall decrease in Aβ. We hypothesize that this regional discrepancy may reflect differences in regional vulnerability to Aβ accumulation or variations in Aβ clearance mechanisms. Additionally, we employed an anti‑Aβ1‑42 antibody validated for detecting monomers, oligomers, and fibrils, providing a comprehensive assessment of amyloid pathology. We have updated the discussion to reflect these points.

(13) Line 208: Please give the rationale for the mTOR expression analyses. Since AD pathology is known to dysregulate autophagic pathways and based on our previous findings that calcineurin inhibition can reinitiate autophagy we assessed mTOR expression as a marker of autophagy. Unexpectedly, we observed an increase in phosphorylated mTOR following TAC treatment, suggesting a nuanced interaction with the mTOR pathway. We have updated the discussion to elaborate on this observation and its implications for TAC’s modulation of autophagy.

(14) The authors state that the sub-immunosuppressive 0.1mg/kg TAC treatment is more beneficial. However, in the paper...DOI:10.1002/alz.061984, the authors showed that 3xTG-AD mice treated with higher FK506 (i.e. TAC) dose (1mg/ml) treated mice displayed significant lower levels of Aβ and Tau in all the considered areas as compared to PBS‐treated 3xTg‐AD mice. How are these results compared to the results obtained in this study? Why is the dose of 0.1mg/kg more beneficial? Were there any documented negative side-effects when the higher dose was used? We thank the reviewer for this important comment. While both doses reduced AD pathology, the sub-immunosuppressive dose (0.1 mg/kg) is particularly attractive because it avoids the risks associated with chronic immunosuppression. Although the higher dose (1 mg/kg) effectively reduced Aβ and Tau levels, it produced blood levels consistent with immunosuppression, even though no overt adverse effects were observed. Our findings suggest that if a lower dose is sufficient to reverse AD pathology, it would be preferable for human application to minimize potential side effects associated with immunosuppression.

(15) The images are missing in the original images/gels folders. We apologize for this oversight. All original images and gels have now been uploaded to the designated folder to ensure full transparency and compliance with the journal’s requirements.

Reviewer 2 Report

Comments and Suggestions for Authors

Comments to the Authors

The authors conducted a comprehensive study titled “Sub-Immunosuppressive Tacrolimus Ameliorates Amyloid-Beta and Tau Pathology in 3xTg-AD Mice,” exploring the potential repurposing of tacrolimus as a therapeutic agent for Alzheimer’s disease. Their investigation delves into a series of methodical studies on TAC (FK506), focusing on protein analysis and histological assessments to elucidate the mechanisms that underpin the interactions between tau pathology and Alzheimer’s disease. However, the study could greatly benefit from incorporating demographic factors such as age and gender, enhancing the research findings' breadth and depth.

Here are some comments to improve

1.      The introduction part would be strengthened by providing insight into the mechanisms of how FK506 crosses the blood-brain barrier (BBB), further contextualizing the study's findings.

2.      Regarding the animal group and sample sizes utilized, additional details would be helpful. Notably, the absence of statistical analysis in the blood analysis section is concerning and should be addressed.

3.      The authors reported that “TAC treatment also modulated autophagy pathways, as evidenced by increased levels of p62, Atg5, Atg7, and LC3, indicating an activation of cellular clearance mechanisms.” However, this assertion requires supporting evidence to validate the claims made. Additionally, Figure 6 needs to be reorganized according to the legends to avoid confusion for the reader.

4.      It is also imperative to include information regarding the vehicle control group, as this data is crucial for understanding the experimental design. Lastly, providing detailed information about the LC/MS spectra and the quantification calculations would enhance the transparency and reproducibility of the research findings.

5.      Is there any reason for the variation of Amyloid-B and p-Tau in different regions of hippocampal tissues such as CA1, CA3, and DG?

Author Response

(1) The introduction part would be strengthened by providing insight into the mechanisms of how FK506 crosses the blood-brain barrier (BBB), further contextualizing the study's findings. We thank the reviewer for this suggestion. In the revised introduction, we now highlight that TAC’s lipophilic nature facilitates its passage across the BBB, allowing it to reach central nervous system targets and modulate neuronal signaling. A relevant citation has been added.

(2) Regarding the animal group and sample sizes utilized, additional details would be helpful. Notably, the absence of statistical analysis in the blood analysis section is concerning and should be addressed. We have added additional details regarding the animal groups and sample sizes in the figure legends and Methods section. We also clarified that pooled whole blood samples precluded statistical analysis.

(3) The authors reported that “TAC treatment also modulated autophagy pathways, as evidenced by increased levels of p62, Atg5, Atg7, and LC3, indicating an activation of cellular clearance mechanisms.” However, this assertion requires supporting evidence to validate the claims made. Additionally, Figure 6 needs to be reorganized according to the legends to avoid confusion for the reader. We thank the reviewer for this comment. In response, we have re-added Figure 6, which now provides the supporting evidence for the modulation of autophagy pathways (as shown by increased levels of p62, Atg5, Atg7, and LC3).

(4) It is also imperative to include information regarding the vehicle control group, as this data is crucial for understanding the experimental design. Lastly, providing detailed information about the LC/MS spectra and the quantification calculations would enhance the transparency and reproducibility of the research findings. Prior to the chronic TAC exposure study (Figure 1), an acute TAC exposure experiment was performed where mice received a single dose of TAC (1 mg/kg or 10 mg/kg) or VEH and were euthanized 6 hours later. Whole blood was collected, and since TAC was not detected in the VEH group, this testing was not repeated in the chronic study. This data, along with the LC/MS spectra, has been added as supplementary material.

(5) Is there any reason for the variation of Amyloid-B and p-Tau in different regions of hippocampal tissues such as CA1, CA3, and DG? We thank the reviewer for this insightful comment. The Discussion has been updated to note that, while Aβ reduction was statistically significant only in the CA1 region, combined analysis of the CA1, CA3, and DG regions revealed overall decreases in both phosphorylated-tau and Aβ levels in TAC-treated 3xTg-AD mice. These regional differences may reflect variations in susceptibility to Aβ pathology or clearance mechanisms, or the 14-day treatment may not fully capture changes across all subregions.

Round 2

Reviewer 1 Report

Comments and Suggestions for Authors

I think that the authors have adequately addressed all the comments in the
revised version of the manuscript.